# The Role of Micro Waqf Bank in Women's Micro-Business Empowerment through Islamic Social Finance: Mixed-Method Evidence from Mawaridussalam Indonesia

**Andri Soemitra** \* **, Kusmilawaty and Tri Inda Fadhila Rahma**

Faculty of Islamic Economics and Business, Universitas Islam Negeri Sumatera Utara, Medan 20221, Indonesia; Kusmilawaty@uinsu.ac.id (K.); triindafadhila@uinsu.ac.id (T.I.F.R.)
\* Correspondence: andrisoemitra@uinsu.ac.id

**Abstract:** This study examined the role of the Micro Waqf Bank in empowering women micro-entrepreneurs. This research used a mixed-method with an explanatory design. Quantitatively, this study examines the effect of the joint responsibility system, financing, and mentoring on the growth of the micro-enterprises business. The sample used was 200 women micro-entrepreneurs using a purposive sampling technique. Quantitative data analysis was performed using the Structural Equation Model based on Partial Least Squares (SEM PLS) technique. This research qualitatively analyzes how the Micro Waqf Bank has carried out its role in empowering women, including dealing with crises due to the COVID-19 pandemic. Thematic analysis was used as the qualitative data analysis technique. Informants in this study amounted to 6 informants from MWB Mawaridussalam consisting of Management, Mentors, and female micro-enterprises. The research results quantitatively indicate that there is a positive and significant impact on the joint responsibility system, financing, and mentoring on the development of micro-enterprise members. Mentoring is the most significant variable that impacts the growth of a micro-enterprise business. Qualitatively, the research results show the role of the Micro Waqf Bank in the economic empowerment of women micro-business entrepreneurs by specifically recommending reinforcement of the mentoring function.

**Keywords:** micro waqf bank; women empowerment; Islamic social finance; financing; joint responsibility; mentoring

## 1. Introduction

The economic income gap for women is still a problem that happens throughout the world (Pieters and Klasen 2020). Women are less able to demonstrate power and decision-making both in the household and in the public (Arshad et al. 2021; Rubin et al. 2021). Some conflicts must be handled by women regarding the demands of accomplishing their obligations as wives or daughters who have to share time doing household chores or neglecting household responsibilities to work (Salia et al. 2018). Globally 65% of the world's poor are women (Sikivahan and Ponniah 2019a). Several strategies have been carried out to reduce the gaps that arise including an approach to meeting their basic needs, (Galiè 2018) as well as, economic empowerment programs such as financial inclusion programs (Faulkner 2019; Hendriks 2019). The women's economic empowerment trend in the last three decades is mostly carried out through microfinance institutions (Nyarko 2022; Rahman et al. 2017). Empowering women through microfinance facilities is one of the best instruments to empower communities, especially in rural areas (Subbaiah and Srinivas 2020).

Some studies have shown that empowering women through microfinance institutions is one of the most effective strategies. Stavrevska (2018) strengthens the role of microfinance institutions in improving family welfare and gender equality. Bansal and Singh (2020) stated that women are more developed and socially empowered after obtaining financing through microfinance. Yount et al. (2021) emphasized that microfinance can empower

women without having a detrimental effect. Many other studies have also confirmed the positive impact of microfinance on women's economic empowerment, especially for the poor in rural areas. Microfinance provides poor women with access to income, job creation, access to savings, and improves family living standards (Das 2021; Okunlola et al. 2020; Patel et al. 2018). Along with the increasing trend of Islamic finance, Islamic microfinance is also trusted to carry out the same functions as other microfinance. Several studies have also demonstrated the role of Islamic microfinance in women's economic empowerment (Islam 2021; Rahayu 2020; Rashid and Ejaz 2019).

However, women's empowerment using microfinance cannot be done partially by only providing financing. Any approaches that combine financing with financial literacy, socio-economic training, education, health care, social mobilization, and so on, are needed (Nawaz 2019). So far, no research has been found that studies women's empowerment through financing, joint responsibility system, and mentoring in one comprehensive study. This research was conducted to further examine the role of the Micro Waqf Bank Mawaridussalam Indonesia in empowering women through Islamic social financial instruments.

The main question of this research is whether the women's empowerment program through Islamic social finance at the Micro Waqf Bank is effective in increasing the development of women micro-entrepreneurs. In the Indonesian context, the Micro Waqf Bank is one of the institutions that provide economic empowerment to women in the micro-small business sector. Micro Waqf Bank is an effective waqf-based institution with funding sources coming from individual, institutional, and corporate donations, with a focus on financing small businesses with Islamic scheme financing. Quantitatively, this study examines the impact of three main empowerment factors, namely the system of joint responsibility, financing, and mentoring on the growth of women micro entrepreneurs' businesses. Qualitatively, the research examines the role of Micro Waqf Bank in empowering micro-enterprises business, including in handling business in the crisis due to the COVID-19 pandemic.

This research is urgent for many reasons. First, the micro-financing scheme through the Micro Waqf Bank has been running since 2017 in Indonesia. An evaluation of the role of micro-financing through the productive waqf program at the Micro Waqf Bank is crucial to do by focusing on one of the Micro Waqf Bank. Second, the fact that almost all members of Micro Waqf Banks in Indonesia are women. An evaluation of several important factors that influence the growth of women micro entrepreneurs' business needs to be carried out. This research will become a stepping stone for making strategies and policies to increase women's empowerment in the future. Third, this research is one of the empirical evidence of the effectiveness of the role of Islamic social finance in women's micro-enterprise empowerment programs in Indonesia.

## 2. Theoretical Background and Previous Studies

### 2.1. Microfinance and Micro Waqf Bank

Microfinance provides opportunities for women to improve their standard of living through financing for their micro-enterprises (Tariq 2020; Tariq and Bhat 2020; Zafarullah and Nawaz 2019). Microfinance mainly refers to the provision of microfinance loans, access to savings for the poor, micro insurance, and other microfinance services targeting low-income customers (Bachisse and El Filali 2018; Kaushal et al. 2021). There are different types of microfinance in the world (Pati 2019). One that is present in the Islamic world is Islamic microfinance. Islam (2021) shows that Islamic microfinance has benefited rural women and empowered them. Other researches have also shown the role of Islamic microfinance in poverty alleviation programs and community economic empowerment (Abbas and Shirazi 2015; Abdul-Majeed Alaro and Alalubosa 2019; Abdullahi and Othman 2021; Begum et al. 2019; Islam 2021; Rahayu 2020; Zitouni and Ben Jedidia 2022).

There are several models of Islamic microfinance in Indonesia (Ascarya et al. 2022a). Micro Waqf Bank (MWB) is one of the most popular microfinance institutions in Indonesia. MWB is an Islamic microfinance institution initiated by the Financial Services Authority in

collaboration with the Business Incubation Center and the Amil Zakat Institution which was opened in May 2018. The term bank does not mean that this financial institution is in the form of a bank. Institutionally, the Micro Waqf Bank is a cooperative legal entity and is included in the category of Sharia Microfinance Institutions (Ascarya et al. 2022b). The establishment of MWB aims to provide access to financing for small communities that do not yet have access to formal financial institutions. MWB focuses on financing micro-business entrepreneurs who are domiciled in the Islamic boarding school area. The purpose of the establishment of the Micro Waqf Bank is to reduce the economic inequality of the people through the management of the Waqf Fund. There are three main elements in the creation of MWB, namely, cash waqf donors, Islamic boarding schools, and communities around the rural area that want to improve their welfare.

The operation of MWB has special characteristics that distinguish it from other financial institutions. First, MWB is a non-deposit financial institution because MWB does not accept deposits from the public. This is because all operational sources are based on Islamic social funds. Second, MWB manages funds based on Sharia principles. Third, its members come from poor communities around its rural areas and have the potential to improve themselves. According to data by 2022, there are 62 MWB throughout Indonesia. The cumulative financing amount reaches USD 5,850,000, the amount of outstanding financing is USD 950,000, the number of cumulative customers reached 52.8 thousand people, the number of outstanding customers was 14 thousand people, and 4.4 thousand assisted groups.

*2.2. Women's Economic Empowerments*

The empowerment concept refers to efforts to increase capacity, especially for vulnerable groups (Sawe 2021). Empowerment is carried out in such a way that these weak groups have certain abilities, for example, to meet their basic needs, have freedom, can obtain resources that enable them to effectively increase their income to achieve the goods and services they need, can participate in decision-making regarding their case, and can participate in development. Economic empowerment refers to empowerment to increase the community's ability to meet their basic needs and gain equal access (Al-Dajani and Marlow 2013; Parwez and Patel 2022; Shuja et al. 2020).

Empowering women is included in the fifth sustainable development goal (SDG5) known as gender equality (Atahau et al. 2021). Women's economic empowerment refers to women's efforts to increase their ability to be equally empowered as men, especially in terms of access to productive resources, to be able to participate in the utilization of these productive resources, have the same control over productive resources as men, have an equal distribution from the use of existing productive resources, including national economic development output, and have equal financial literacy (Alloatti 2019). Women's economic empowerment is present as one of the efforts for gender equality, poverty alleviation, and inclusive economic growth (Hendriks 2019; Huis et al. 2019; Lee and Huruta 2022).

One medium to overcome the problem of women's empowerment is to use microfinance institutions. Women have had limitations in accessing business capital (Maurya and Mohanty 2019). The use of microfinance is seen as effective because it increases women's capacity through stages of empowerment, starting from the breakdown of their position, collusive arrangements, quasi-empowerment, positive freedom, and empowerment (Ajwani-Ramchandani 2017). The qualified skills of the managers of microfinance institutions also affect the success of paying out financing to micro-enterprises (Nkamnebe and Idemobi 2011). Building entrepreneurial capacity for women serves as an effective tool for more holistic women's empowerment (Khan et al. 2022). Microfinance institutions need to pay more attention to the condition of their members who receive financing by increasing their capacities (Kassim and Rahman 2018).

Eswarappa (2020) shows that women who are part of Community-based Organizations have a role to play in implementing development programs in their area and have a role in empowering women. Microfinance helps empower women from poor households.

Microfinance has the prospect of creating important engagements in the gender equality category, promoting sustainable sources of income, and enhancing the functioning of the environment reserved for women (Tiwari et al. 2021).

### 2.3. Micro Business Growth and Business Situations during the COVID-19 Pandemic

Micro-enterprises can be the answer to job creation for the lower-class society (Sarmah et al. 2021). Microbusiness entrepreneurs can be expected to develop and progress to the next level of business i.e., small and medium under the correct strategies and policies (Quinn et al. 2014). However, there are several of obstacles encountered by micro-business entrepreneurs in developing their businesses, including: (1) Low income levels causing in a circle of lack of income and capital in running their business; (2) the lack of human resources for micro-enterprises in organizing a sound business; (3) the micro-business entrepreneurs limitations in accessing sources of capital, market information, appropriate technology, marketing, and other production factors; (4) uncertain business location; (5) micro-business entrepreneurs do not have legality; (6) legal and regulatory issues are still not able to significantly encourage micro-enterprises; (7) the low quality of products and services produced by micro-business entrepreneurs; (8) procedures in starting a business that are still not easily understood; (9) government programs to encourage micro-enterprise growth are still limited; (10) production and management costs are still high; (11) lack of coordination between institutions that manage micro-business entrepreneurs (Mthimkhulu and Aziakpono 2015; Gherhes et al. 2016; Mittal and Raman 2021; Sanu and Anjum 2021).

The business obstacles encountered by business entrepreneurs can generally be categorized into two main causes, namely internal and external factors. External factors that hinder the development of micro-enterprises include: (a) Limited access to business financing; (b) high infrastructure costs; and (c) inefficient bureaucratic services. The obstacles to the development of micro-enterprises internally, among others (a) institutions and human resources, (b) marketing, and technology, and (c) intellectual capital (Candra et al. 2021; Prijadi et al. 2020; Susan 2020; Tulus Tahi Hamonangan Tambunan 2011). Therefore, serious efforts are needed to overcome the obstacles so that micro-business entrepreneurs can develop and advance to the next level.

The strategy that allows micro-business entrepreneurs to develop both during normal times and during times of crisis is determined by their efforts to overcome the obstacles (Khan 2022). Some strategies can be applied to overcome internal obstacles such as upgrading human resources, providing affordable raw materials, increasing target markets and marketing strategies, and increasing quality and production capacity using appropriate technology. While externally, some strategies are needed to eliminate external obstacles, namely providing access to capital, friendly licensing and infrastructure to micro-enterprises, and providing policies that are conducive to the growth of micro-enterprises. Including strategies to empower women from a gender perspective (Orser et al. 2021).

The COVID-19 pandemic is making it more difficult for micro-entrepreneurs to grow (Sahoo and Thakur 2022). During the COVID-19 pandemic, micro-business entrepreneurs had movement restrictions. It made it difficult for them to meet face-to-face with buyers. The decline in offline sales volume is unavoidable. The most prominent strategy in overcoming obstacles to business growth during the pandemic is the transformation from manual transactions to virtual transactions. Risk mitigation is also an option to survive during a pandemic. The use of technology media and risk mitigation are the keys to distinguishing business models before and after the COVID-19 pandemic (Afshan et al. 2021; Mustafa et al. 2021; Rahman et al. 2022; Yeon et al. 2022).

### 2.4. Joint Responsibility System

The joint responsibility system was initiated by Grameen Bank, which was founded by Muhammad Yunus in Bangladesh (Hassan and Renteria-Guerrero 1997). The joint responsibility system is mainly related to the new approach to channeling funds, especially to poor micro-enterprises that do not have sufficient funds to start their business. The joint

responsibility system is a reciprocal obligation between a group of members who have a general obligation to pay for the financing. If one or more group members fail to pay the financing capital, all group members are responsible for paying off the capital financing. Having a shared responsibility system in group-based financing is very helpful in building relationships and trust among members (Kong et al. 2015; Mukherjee and Bhattacharya 2015; Rathore 2017).

The implementation of the joint responsibility system has a positive impact on women's empowerment groups. Patel and Patel (2021) explained that in the Indian context, women were significantly empowered from an economic perspective after joining Self Help Groups (SHGs), a non-governmental organization. The pattern of providing loans for SHGs members is providing loans to the poor and marginalized members of SHGs in rotation.

Yaron et al. (2018) proved that the pattern of rotating loans for SHGs members was developed. The success of this group-based rotating loan pattern as an instrument of women's economic empowerment is also strengthened by the research of Wulandari and Kassim (2016); Ancel et al. (2019); Subrahmanyam and Santosh (2019); Sikivahan and Ponniah (2019b); Radhaakrishnan et al. (2019); Gordon (2020); Manikandan and Muthumeenakshi (2020); Gogoi (2020); Manikandan and Muthumeenakshi (2020). The joint responsibility system has been studied as one of the financing mitigation strategies in Islamic microfinance (Cameron et al. 2021). Thus, this research will predict:

**H1.** *There is a positive impact of the joint responsibility system on the growth of women micro-entrepreneur businesses.*

### 2.5. Microfinancing

Microfinancing is an activity of allocating funds by financial institutions to micro-business entrepreneurs for business development or the provision of business infrastructure. Microfinance is channeled to business entrepreneurs who generally work as micro traders, or run micro-scale businesses. The criteria for a micro-scale business in the Indonesian context is to have a net worth of at most USD 571,429 or maximum annual sales of USD 21,428,572 (Tambunan 2009).

The Islamic microfinance scheme is carried out in compliance with sharia principles. Islamic microfinance avoids interest-based financing and all types of businesses that use illicit products and services. The financing scheme of Islamic microfinance uses sharia contracts that are under customer needs, such as murabaha (sale and purchase contracts), mudharabah (profit-sharing contracts), or qardh al hasan (interest-free loan contracts). The financing scheme of Islamic microfinance can use the individual financing model with or without collateral and can also use the group financing model without collateral based on a joint responsibility system (Ascarya and Sakti 2022; Haneef et al. 2015; Hassan and Saleem 2017; Kakembo 2021; Wulandari 2019).

Hussain et al. (2019) showed that persistent financial exclusion, gender discrimination, and conservative religious values harm women's empowerment. This study offers evidence and supports theories about the positive impact of microcredit on women's poverty alleviation. Likewise, research by Abubecker et al. (2019) confirms that financial inclusion is one of the important instruments in women's economic empowerment. The empowerment of business entrepreneurs will be more successful if the business capabilities of business actors are supported by business financiers (Nieuwenhuizen and Kroon 2002). Some studies have shown the positive influence of microfinance to improve the welfare of microbusiness entrepreneurs, among others (Mariyono 2019; Nkamnebe and Idemobi 2011; Obamuyi 2009). This research will predict:

**H2.** *There is a positive impact of microfinancing on the growth of women micro-entrepreneur businesses.*

### 2.6. Entrepreneurial Mentoring

Mentoring is an activity that is implemented by the activity companion in the community action plan. The task of the companion is to act as a stimulus, catalyst, and driver for a

certain activity in the community. Mentoring is carried out to build the independence of members of the receiving community group so that they are free from external dependence. Mentoring is an important process to empower communities, which in turn leads to efforts to improve their welfare (Barrett 2006).

Mentoring is a general strategy used to improve the quality of human resources to gain the ability to identify problems and find alternative solutions for themselves. The mentoring process is very important in the empowerment process. Effective business assistance is required in reducing the inability of business entrepreneurs to run their businesses properly (McKevitt and Marshall 2015). In the context of micro-enterprises, support is generally given to increasing community capacity, beginning from starting a business, understanding the production process, mastering financial accounting, reporting, marketing, appraisal, and continuing to build business capacity.

Several studies confirm that business mentoring has a positive impact on business sustainability. Even though they are different, there is a relationship between mentoring and the results of business mentoring (skill transfer results, knowledge transfer results, entrepreneurial resilience results, and business results) (Kunaka and Moos 2019; St-Jean et al. 2018). Sharafizad (2018) shows the need for mentoring to support the success of women small business entrepreneurs. Rickard and Rickard (2009) emphasized the need to open up opportunities for electronic mentoring to expand opportunities for small business entrepreneurs to get business assistance. Ugwu-oju et al. (2020) emphasized that increasing the management capacity of business entrepreneurs is important for the effective use of the business financing allocation. Ting et al. (2017) emphasized that the sincerity of the mentor to improve the ability of the companion was more important than the quality and skill of the mentor. The research will also predict:

**H3.** *There is a positive impact of Entrepreneurial Mentoring on the growth of women micro-entrepreneur businesses.*

### 3. Methodology

This research was conducted in Mawaridussalam, North Sumatra, Indonesia in February 2022. This study used a Mixed method with Explanatory Design (Creswell and Clark 2007; Creswell and Creswell 2018; Taguchi 2018). In the first stage, quantitative methods were used to analyze whether there was an effect of the joint responsibility, financing, and mentoring system that has been implemented by MWB Mawaridussalam on the business development of women micro-business entrepreneurs. In the second stage, a qualitative approach was used to analyze the MWB pattern in empowering women micro-business entrepreneurs based on Islamic social funds, especially during the COVID-19 pandemic.

*3.1. Quantitative Approach*

The population of this research is women micro-business entrepreneurs who are members of the Mawaridussalam MWB totaling 426 people. According to Hair et al. (2010) for quantitative research, a sample of 100 to 200 is included in the medium category. The sample-set in this study amounted to 200 women micro-business entrepreneurs. Data were obtained by distributing questionnaires to the samples. Samples are selected by purposive sampling technique (Cassel and Lyberg 2001) under the criteria of having been a member of the MWB for at least 1 year; still actively receiving the Mawaridussalam MWB empowerment program, and using financing from MWB for micro-business purposes. The data analysis technique used SEM PLS (Hair et al. 2019a). SmartPLS software version 3.2.9. (University of South Alabama, Alabama, AL, USA), is used to analyze the impact of the joint responsibility system (X1), Financing (X2), and Mentoring (X3) on the Growth of Women's Micro Enterprises (Y) at MWB Mawaridussalam.

*3.2. Qualitative Approach*

For the qualitative research, interviews, observations, and document studies were conducted on the MWB Management and some female micro-entrepreneurs who were

members of the MWB. The qualitative approach uses an interactive model starting from the stages of data collection, data condensation, data display, and conclusion data. This model includes the steps of establishing a code, preparing interview transcripts or documents; preliminary data exploration on the results of interview transcripts; filtering and assigning data coding according to the theme; linking relationships between themes, elaborating a consistent set of questions, propositions, and generalizations found in the database; compare generalizations with formal knowledge; and generate a narrative (Creswell and Creswell 2018; Miles et al. 2014; Perreault 2011).

The data analysis technique used is thematic analysis. This analysis is used because thematic analysis is the accurate method to identify, analyze, and report the patterns (themes) that exist in the data, both to interpret simple and complex meanings (Crawley 2020; Ozuem et al. 2022). Informants in this study amounted to 6 people from MWB Mawaridussalam consisting of Management, Assistant Managers, and the women micro-business entrepreneurs MWB members. Profile of the interviewees can be seen in Table 1. The number of informants 6 people is based on the suggestion of Morse and Richards (2002) as well as another qualitative research, i.e., (Hapsari et al. 2021) which states that an in-depth study with a small number of informants is sufficient to achieve research objectives.

**Table 1.** Profile of the interviewees.

| No | Position in MWB | Code |
|---|---|---|
| 1. | Board of Directors | I1 |
| 2. | Manager | I2 |
| 3. | Supervisor | I3 |
| 4. | Mentor | I4 |
| 5. | Mentor | I5 |
| 6. | Member | I6 |

The key information that needs to be extracted from informants is about how the pattern of empowerment is carried out by MWB, including during the COVID-19 pandemic. The thematic analysis in this research operationally uses the analysis of Miles and Huberman (1994) selecting data, reducing data, and coding data based on similar themes.

## 4. Results and Discussion

### 4.1. Characteristics of Respondents

In terms of length of time running a business, all respondents have run a business within a certain period. (see Table 2).

**Table 2.** Characteristics of respondents based on length of running a business.

| Description | Amount |
|---|---|
| <1 year | 2 |
| 1–2 years | 74 |
| 3–4 years | 106 |
| >5 years | 9 |
| Total | 200 |

Source: Research Questionnaires 2022.

The results of the research show that respondents are dominated by business durations of 3–4 years as many as 106 female micro-enterprises, followed by 74 women micro-entrepreneurs within 1–2 years. Nine women micro-entrepreneurs have run a business in a span of >5 years. There are only two respondents who have less than 1 year of running the business.

The characteristics of respondents based on initial capital starting the business can be seen in Table 3.

**Table 3.** Characteristics of respondents based on initial capital starting the business.

| Description | Amount | Percentage |
|---|---|---|
| Rp. 500,000 (USD 36) | 6 | 3 |
| Rp. 500,000 (USD 36)–Rp. 1,000,000 (USD 71) | 32 | 16 |
| Rp. 1,000,000 (USD 71)–Rp. 1,500,000 (USD 107) | 62 | 31 |
| >Rp. 2,000,000 (USD 145) | 100 | 50 |
| Total | 200 | 100 |

Source: Research Questionnaires 2022.

From the amount of initial capital used to start a business, the respondents in this study were mostly more than Rp. 2,000,000 (USD 145) i.e., as many as 100 people followed by Rp. 1,000,000 (USD 71)–Rp. 1, 500,000 (USD 107) as many as 62 people, starting a business with a capital of Rp. 500,000 (USD 36)–Rp. 1,000,000 (USD 71) as many as 32 people, and Rp. 500,000 (USD 36) as many as 6 respondents. This condition indicates that the respondents started a business by preparing a certain amount of initial capital which is included in the category of micro business capital.

The characteristics of respondents based on financing amount from MWB can be seen in Table 4.

**Table 4.** Characteristics of respondents based on financing amount from MWB.

| Description | Amount | Percentage |
|---|---|---|
| Rp. 500,000 (USD 36) | 0 | 0 |
| Rp. 1,000,000 (USD 71) | 32 | 16 |
| Rp. 1,500,000 (USD 107) | 62 | 31 |
| Rp. 2,000,000 (USD 145) | 106 | 53 |
| Total | 200 | 100 |

Source: Research Questionnaires 2022.

Data on the amount of financing distributed by MWB to respondents of women micro-business entrepreneurs obtained that the financing results were dominated by the financing of Rp. 2,000,000 (USD 145) as many as 106 people, then followed by Rp. 1,500,000 (USD 107) for 62 people, and Rp. 1,000,000 (USD 71) to as many as 32 people. This data show that women micro-business respondents have obtained micro-financing in the maximum financing scheme provided by MWB, which is Rp. 2,000,000 (USD 145). All respondents obtain financing to finance their businesses using a joint responsibility system. Based on the amount of financing with a joint responsibility system that is distributed to all women, micro-business entrepreneurs are included in the micro-business sector.

*4.2. Quantitative Analysis of the Effect of Joint Responsibility System, Financing, and Mentoring on Women's Micro Business Growth*

4.2.1. Factors and Indicators in the Model

This study tested the hypothesis using SEM (structural equation modeling) based on partial least square (PLS). This study carried out two stages of quantitative testing, namely, the first stage was testing the measurement model, namely testing the validity and reliability of the constructs (variables) of each indicator. The second stage was to conduct a structural model test to know whether there is an impact between the variables or a correlation between the constructs.

This quantitative research initially consisted of four variables with 29 indicators. The independent variable is the growth of women's micro-enterprises after becoming members of the Micro Waqf Bank at the Mawaridussalam (Y) consisting of eight indicators i.e., (1) increase in business turnover, (2) increase in the number of buyers, (3) increase in profit, (4) increase in business income, (5) increase in business assets, (6) business

development, (7) business expansion, (8) improvement in meeting the basic economic needs. The dependent variable is the Joint Responsibility System (X1) consisting of seven indicators i.e., (1) group financing, (2) convenience, (3) benefits, (4) access to capital, (5) shared responsibility, (6) honesty and discipline in installment payments, (7) mutual respect between members; microfinancing (X2) consisted of seven indicators i.e., (1) easy initial requirements, (2) financing to increase business capital, (3) low financing administration costs, (4) financing sufficient for business needs, (5) financing used for business activities, (6) affordable installment payments, (7) the repayment period is not burdensome; and mentoring (X3) consisted of seven indicators i.e., (1) mentors assist members' business development, (2) mentors provide information on managing products and services with appropriate technology, (3) mentors assist in business bookkeeping, (4) able to prepare financial reports correctly, (5) implement marketing techniques taught by mentors, (6) get guidance and solutions in overcoming business problems, (7) mentoring and monitoring motivate members to always develop their businesses. The questionnaire is designed according to the question indicators (factors items) to be able to measure the latent variables. Respondents answered questions according to ranking on a 1–5 point Likert scale.

### 4.2.2. Confirmatory Factor Analysis

Factor analysis was conducted to validate the questionnaire to members of micro business actors. The validity of the factor structure is verified by referring to the factor loading. This research uses the considerations of Kaiser (1974) who stated that a minimum value of 0.5 is still considered, although according to Kaiser (1974) and Csath et al. (2018) a value of >0.8 is considered very good in factor analysis. The SEM result of this research can be seen in Figure 1.

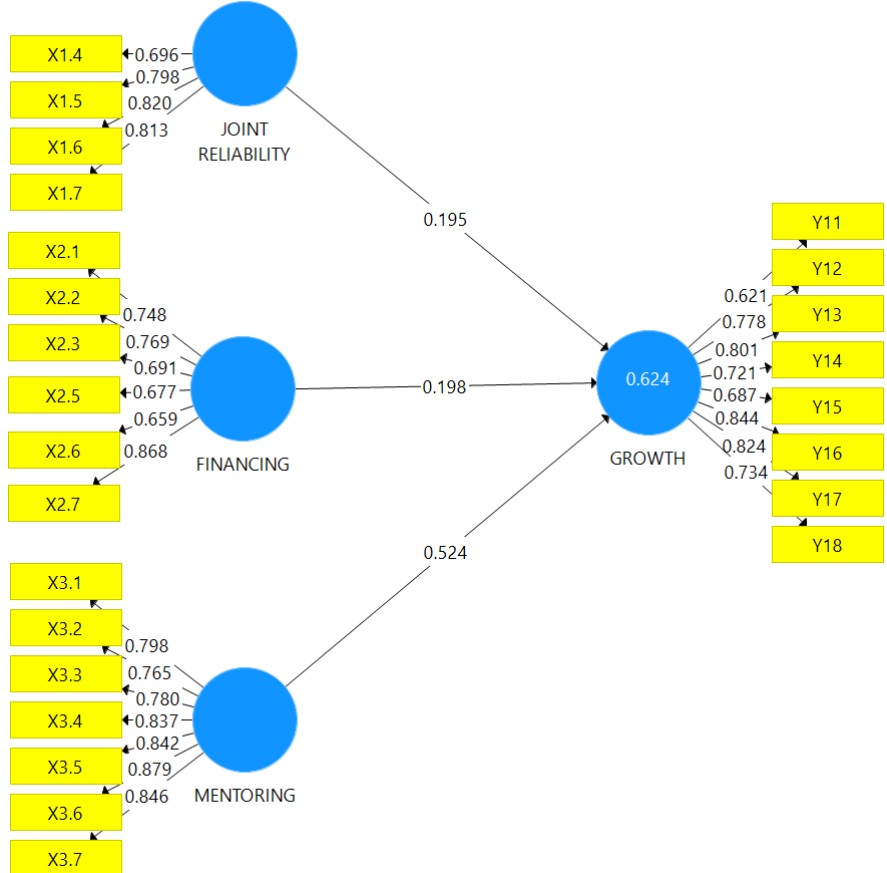

**Figure 1.** SEM result (convergent validity).

The results of the validity test of the questionnaire test left 25 items above the loading factor of 0.5. Therefore, each factor has been supported to be used in the factor analysis because all factor loadings are above 0.5. After the validation of the factors and indicators of the factors, the PLS-SEM analysis was carried out.

### 4.2.3. Assessment of Outer (Measurement) Model

Evaluation of the measurement model uses convergent and discriminant validity of the construct recommended by Hair et al. (2021); Leguina (2015). The assessment result of outer (measurement) model of this research can be seen in Table 5.

**Table 5.** Reliability and validity test results.

|  | Cronbach's Alpha | Rho_A | Composite Reliability | AVE |
|---|---|---|---|---|
| Financing | 0.831 | 0.842 | 0.877 | 0.564 |
| Growth | 0.890 | 0.899 | 0.913 | 0.569 |
| Join Liability | 0.789 | 0.801 | 0.863 | 0.613 |
| Mentoring | 0.920 | 0.924 | 0.936 | 0.675 |

Source: author based on SEM PLS output.

In testing the reliability of all variables in this study, the value of Cronbach's alpha > 0.6 and the value of composite reliability > 0.7, and the AVE value above 0.5 which means that the reliability and validity of this study are satisfied.

### 4.2.4. Assessment of Inner (Structural) Model

The structural model was analyzed using the bootstrapping procedure suggested by Hair et al. (2014). The t-statistical test is a way to observe the significant effect of the prediction model in the inner model or in other words to see the results of the hypotheses in this study. The t-statistical test is declared influential if the *p*-value < 0.05 or the t-statistical value > 1.65251. The bootstrapping results (Table 6) show that 3 (three) constructs (variables) namely the Joint Responsibility System (X1), Financing (X2), and Mentoring (X3) have a positive and significant influence on the Growth of Women's Micro Enterprises (Y). Thus, the results of this research are in line with previous research. The results of this study also indicate that the Mentoring variable is the most significant factor influencing the development of micro-enterprises. This shows that Mentoring needs greater attention to becoming the basis for improving the development of micro-customer businesses in the policy-making process in the future.

**Table 6.** Bootstrapping results.

| Hypothesis | Original Sample (O) | Sample Mean (M) | Standard Deviation (STDEV) | t Statistics (∣O/STERR∣) | *p* Values | Hypothesis Test Results | $R^2$ |
|---|---|---|---|---|---|---|---|
| There is a positive impact of Micro Financing on the Growth of Micro Enterprises | 0.198 | 0.198 | 0.069 | 2.857 | 0.004 | Accepted | |
| There is a positive impact of the Joint responsibility System on the Growth of Micro Enterprises | 0.195 | 0.197 | 0.054 | 3.633 | 0.000 | Accepted | |
| There is a positive impact of Mentoring on the Growth of Micro Enterprises | 0.524 | 0.525 | 0.072 | 7.289 | 0.000 | Accepted | 0.624 |

Source: author based on SEM PLS output.

$R^2$ (R-square) measures the variance described in each endogenous construct and is used to see the magnitude of the influence of all independent variables on the dependent variable (Hair et al. 2019b). $R^2$ ranges from 0 to 1, with higher values indicating greater explanatory power. As a guideline, if the $R^2$ value of 0.75 is substantial; 0.50 means moderate, and 0.25 means weak (Hair et al. 2011; Henseler et al. 2009). The value of $R^2$ (R-square) in this study was 0.624 (between > 0.50 < 0.75), in the moderate category. This means that all independent variables i.e., Joint Responsibility System (X1), Financing (X2), and Mentoring (X3) have moderate impact on the dependent variable i.e., the Growth of Women's Micro Enterprises (Y). However, when using Cohen (1988) criteria if an $R^2$ value of greater than 0.40 then this model is considered quite substantial.

*4.3. Qualitative Analysis of Micro Business Growth Patterns at MWB Mawaridussalam*

4.3.1. The MWB Financing Role to Support Family Income

The results of the interview with the Management provided information (I1) that the MWB Mawaridussalam was established in 2018 and was inaugurated by the President of the Republic of Indonesia. In 2022, MWB Mawaridussalam distributed capital financing for micro-business entrepreneurs to 426 members totaling Rp. 689 million. Furthermore, based on the results of interviews with female customers of micro-enterprises, it was found that the business capital financing provided by MWB Mawaridussalam was only able to become a supporting fund in helping to increase family income. This is as stated by the informant as follows: "Alhamdulillah, the capital financing assistance received from MWB Mawaridussalam is indeed small. But that's enough to help small business capital. Income from small businesses can be used to help increase family income" (I6). It must be admitted that MSME capital financing has not been able to become the main source of income because the amount of financing is small and it targets micro sector businesses that also provide micro income.

4.3.2. Empowerment Roles of Women Micro Business Entrepreneurs by MWB

Based on the subsequent interviews, it was found that until now the distribution of MWB Mawaridussalam has 0% Non-Performing Finance. This is following the statement of the informant (I2): "So far, MWB Mawaridussalam has succeeded in keeping customers from fulfilling their obligation to pay for the capital financing they received. All customers are obedient to pay. MWB Mawaridussalam managed to keep the Non-Performing Finance (NPF) at 0%." The commitment to keep the NPF at 0% is based on the source of the MWB funds that do not come from members' deposit funds. MWB does not accept savings from the public. The source of financing for MWB comes from Islamic social funds which are managed on a revolving basis by its members.

MWB Mawaridussalam has a unique pattern of economic empowerment for women micro-business actors. Based on the results of interviews with management informants (I1, I2, I3), assistants (I4 and I5), and customers (I6), information was obtained that the process of empowering micro business actors had started from the initial selection by preparing the capacity building for prospective members. According to the informant, mentoring is the main strength of MWB in empowering women micro-business entrepreneurs. The stages of empowerment carried out by MWB Mawaridussalam can be described as follows:

1.  Prospective members participate in the selection of prospective customers through the Halaqah "Compulsory Group Training" (LWK) for "Community Business Groups around Islamic Boarding Schools" (KUMPI). This selection activity took place for five consecutive days with the material of discipline, cohesiveness, solidarity, and the courage to try. This Halaqah must be followed by every prospective member. Discipline in following this Halaqah is considered as collateral for MWB to provide group financing or cancel it.
2.  MWB formed a group with the name "Halaqoh Weekly" (HALMI) consisting of three to five groups. Each group consists of five people. These groups are then trained

regularly once a week. The coaching materials provided by MWB are related to solidarity, business commitment, and togetherness.

3. MWB disbursed business financing to groups that passed the selection at the first meeting of HALMI activities. The management of financing capital is carried out jointly and severally by the group. Each group member is responsible for managing the funds and ensuring that the financing is used for group members' micro-enterprise needs.

4. At the next meeting, the MWB provides assistance and guidance through various activities to enrich members' capacity-building materials such as business development techniques, and household economic management.

5. The MWB holds regular HALMI meetings. It is at the weekly meeting that members pay the installments of the business capital financing. The customer is required to pay off the capital financing with an additional low margin of only 3% based on Murabaha contract. This weekly meeting is very strategic because apart from being a medium for paying installments, it is also a medium for MWB to monitor business and increase the business capacity of members.

6. At regular HALMI meetings, MWB continues to carry out a coaching and mentoring process including presenting competent practitioners to educate group members to imitate and practice the best business models.

The success of the empowerment of the Mawaridussalam MWB is supported by the data from the questionnaire distribution which shows an increase in the business profit of women micro-business entrepreneurs after receiving financing and mentoring from the Mawaridussalam MWB. These data can be seen in Table 7.

**Table 7.** Operating profit before and after obtaining financing and mentoring.

| No | Average Monthly Operating Profit before Receiving Financing from MWB | Amount | Percentage |
|----|---------------------------------------------------------------------|--------|------------|
| 1 | <Rp. 500,000 (USD 36) | 62 | 31 |
|  | Rp. 500,000–Rp. 1,000,000 ((USD 71)) | 76 | 38 |
|  | Rp. 1,000,000–Rp. 1,500,000 (USD 107) | 38 | 19 |
|  | >Rp. 1,500,000 | 26 | 13 |
|  | Total | 200 | 100 |
| | **Average operating profit after receiving financing from MWB every month** | **Amount** | **Percentage** |
| 2 | <Rp. 500,000 (USD 36) | 0 | 0 |
|  | Rp. 500,000–Rp. 1,000,000 (USD 71) | 44 | 22 |
|  | Rp. 2,000,000 (USD 145)–Rp. 3,000,000 | 100 | 50 |
|  | >Rp. 3,000,000 (USD 214) | 56 | 28 |
|  | Total | 200 | 100 |

Source: 2022 research questionnaire.

The average operating profit of female micro-enterprises who became respondents in this research each month before receiving financing from the respondents' MWB varied. There are 62 women micro-business respondents who earn an average profit of < Rp. 500,000 (USD 36). There are 76 respondents of women micro-business entrepreneurs who earn an average profit of Rp. 500,000 (USD 36)–Rp. 1,000,000 (USD 71). Furthermore, there are 38 women micro-enterprise respondents who earn an average profit of Rp. 1.000.000 (USD 71)–Rp. 1,500,000 (USD 107) and 26 respondents of female micro-enterprises who earn an average profit of > Rp. 1,500,000 (USD 107).

The results showed that there was an increase in the average income of the respondents. There are no more women micro-enterprise respondents who earn an average profit of < Rp. 500,000 (USD 36). The profit range of all respondents has increased showing higher profits. There are as many as 44 women micro-business entrepreneurs who earn an average profit of Rp. 1.000.000 (USD 71)–Rp. 1,500,000 (USD 107). The majority of respondents get an average operating profit after receiving financing from MWB every month of as

much as Rp. 2,000,000 (USD 145)–Rp. 3,000,000 (USD 214) totaling 100 respondents, and 56 respondents getting an average operating profit of > Rp. 3,000,000 (USD 214). These data are evidence that the empowerment program played by the Micro Waqf Bank is effective in increasing the average profit of women micro-business respondents.

4.3.3. Micro Business Empowerment Strategy during the COVID-19 pandemic

Based on the results of in-depth interviews, it was found that during the COVID-19 pandemic, the micro-business conditions of MWB Mawaridussalam members experienced tough pressure. The type of business of the Mawaridussalam MWB members consists of 40% of the laundry business, boarding houses, and hotel services because the MWB is located around Kualanamu International Airport. The closure of the airport, which was followed by restrictions on movement, caused the micro-enterprise sector to suffer the worst impact. The composition of MWB Mawaridussalam members, furthermore, 35% in the home industry consists of the production of chips, salted eggs, and herbal drinks; 20% percent are small traders who sell snacks for children, and the remaining 5% are farm laborers. According to the informant (I6), the Government's restrictions on people's movement caused the income of the micro-enterprises that were run to experience a severe decline.

During the COVID-19 pandemic, Mawaridussalam MWB faced specific obstacles in mentoring its members. According to the Informants (I4 and I5), the difficulty in conducting routine mentoring through the Weekly Halaqah Meeting (HALMI) caused the mentoring process to be ineffective. Many of their members do not have android phones, so it is difficult to communicate via social media. Whereas communication via social media is an alternative route for MWB Mawaridussalam to provide mentoring and assistance to their members. The manager of the Mawaridussalam MWB has some strategies to get around the constraints of offline and online communication limitations due to restrictions on movement during the COVID-19 pandemic. According to informants (I4 and I5), the mentoring from MWB Mawaridussalam has taken several tactics, including actively promoting MWB member products through social media. MWB Mawaridussalam also continues to hold weekly meetings amid restrictions on movement by strictly enforcing health protocols.

The management of the Mawaridussalam MWB also has several strategies and policies so that women micro-business members are still able to maintain business continuity amidst the economic pressures affected by the COVID-19 pandemic. Some strategies implemented by MWB so that members are still able to pay installments for capital financing are explained as follows:

1. MWB Mawaridussalam provides business education in dealing with special situations of pandemic conditions for members of micro-enterprises.
2. The management of the Mawaridussalam MWB registers members of female micro-enterprises in the government's social assistance program to be able to survive during the crisis due to the COVID-19 pandemic.
3. The MWB management provides relaxation of payments to their members under government regulations.
4. The MWB encourages its members to open group savings for each joint business group in the cooperative so that each member still has a shared savings account to be a source of installment payments if individual members experience problems with installment payments.
5. The MWB strengthens the implementation of the joint responsibility system for its members. The strengthening of the joint responsibility system is carried out to increase the togetherness of members in dealing with crises and jointly overcome business obstacles faced by members of groups based on joint responsibility.

Thus, the results of this research quantitatively confirm the results of previous research which show that there is an impact of the Joint Responsibility System on Micro Business Growth (Kong et al. 2015; Wulandari and Kassim 2016; Ancel et al. 2019). There is also

the impact of Micro Financing on Micro Business Growth (Mariyono 2019; Nkamnebe and Idemobi 2011; Obamuyi 2009). This research also strengthens the finding that there is an effect of Mentoring on Micro Business growth (Kunaka and Moos 2019; St-Jean et al. 2018; Ugwu-oju et al. 2020).

This study qualitatively confirms previous research that shows the positive contribution of microfinance institutions to women's economic empowerment (Maurya and Mohanty 2019). In the context of this study, the research subjects were women micro-business actors who obtained micro-business capital financing from the Micro Waqf Bank. This research strengthens the argument that women micro-entrepreneurs who receive financing need to get increased capacity-building training and mentoring in running a business so that the empowerment process can run effectively (Kassim and Rahman 2018; Khan et al. 2022). The management of microfinance institutions needs to show their seriousness in assisting microbusiness actors who are members of MWB financing. The persistence and seriousness of mentoring are important factors in supporting the business success of microbusiness entrepreneurs (Nkamnebe and Idemobi 2011).

This research uses the Mawaridussalam MWB as a model for empowering women micro-entrepreneurs using Islamic social funds, which are cheap funds that can be rolled out for micro-business capital financing at low costs. MWB maximizes waqf and alms funds by channeling them into micro-capital financing with a 3% margin under a Murabaha contract scheme. The results of this study have implications for the need for the management of microfinance institutions to not only disburse capital financing to their members. Micro-enterprises need to obtain capacity building and mentoring in managing their businesses. In addition, implementing a joint responsibility system in which each group member is responsible for paying the financing installments is an alternative for collateral. The joint responsibility system can be an effective "collateral" in ensuring that its members complete their obligations to pay installments based on collective group guarantees. The government can also cooperate with microfinance institutions in distributing cheap funds for the economic empowerment of the poor, especially women. Low-cost funds provide an opportunity for customers to pay low fees. Education, experience, and business mentoring obtained from Microfinance Institutions can be very valuable social capital for poor women to increase women's empowerment and contribution, especially in family economic resilience activities.

## 5. Conclusions

The results of this study strengthen previous research which confirmed that microfinance institutions have an important role in empowering women who are not covered by formal financial institutions. Quantitatively, the research confirms that the joint responsibility system, capital financings from microfinance, and mentoring have a positive and significant influence on the development of micro-businesses for women MWB Mawaridussalam members. Qualitatively, the results of this study indicate that microfinance institutions should not only provide business capital financings to their members. Improving the integrity and capacity of their member needs to be a concern. The results of the study show that fostering customer integrity has begun since the selection process to become members and is strengthened by making the joint responsibility system a non-physical "collateral" that ensures members are collectively responsible for repaying capital financing. It is also the responsibility of microfinance institutions to increase customer capacity through regular education and mentoring.

This research, of course, has some limitations. First, this research is a case study using the Mawaridussalam Micro Waqf Bank as the research locus. This case study research is very limited and cannot be generalized. Second, this study focuses on the research theme of empowering female micro-enterprises on three variables, namely the joint responsibility system, micro-financing, and mentoring. Of course, other variables influence the development of the customer's micro-business. Third, the scope of this research is limited to the research of Islamic microfinance institutions in Indonesia. Of course, Islamic

microfinance institutions do not only exist in Indonesia but also in other countries region-
ally and internationally. However, although this research has limitations, this research is
methodologically capable of both quantitatively and qualitatively answering the problem
formulation and contributing to policymakers, microfinance institutions, and the scientific
treasures of microfinance and women's economic empowerment.

Suggested future research includes conducting studies with more cases covering Micro
Waqf Banks throughout Indonesia, examining other variables that affect the development
and growth of micro-businesses at Waqf Banks in Indonesia, and expanding studies on
the development of micro-businesses with wider studies between and among countries,
including regionally, and internationally.

**Author Contributions:** Conceptualization, A.S.; data curation, K. and T.I.F.R.; formal analysis, A.S.;
funding acquisition, A.S.; investigation, A.S.; methodology, A.S.; project administration, T.I.F.R.;
resources, A.S.; software, A.S., K. and T.I.F.R.; supervision, A.S.; validation, K. and T.I.F.R.; writing—
original draft, A.S.; writing—review and editing, A.S. All authors have read and agreed to the
published version of the manuscript.

**Funding:** This research was funded by BOPTN LP2M UIN Sumatera Utara Medan Tahun 2020 Dana
DIPA 2021, grant number Rp. 125,000,000.

**Institutional Review Board Statement:** Not Applicable.

**Informed Consent Statement:** Not Applicable.

**Data Availability Statement:** Not applicable.

**Conflicts of Interest:** The authors declare no conflict of interest.

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
