# Peer review of "The Role of Micro Waqf Bank in Women’s Micro-Business Empowerment through Islamic Social Finance: Mixed-Method Evidence from Mawaridussalam Indonesia"

_economies, doi:10.3390/economies10070157_

Round 1
Reviewer 1 Report
General remarks
1. This research used a mixed-method with an explanatory design, this is a really good point. As mixed methods are not yet trendy, you may indicate a « mixed-method approach » somewhere in the title for a better paper discovery.
3. For un better understanding of the different amounts communicated, please provide the equivalence in USD.
Specific remarks
- Lines 115-116: the same sentence appears twice.
- Line 164: Human Resources replace by human resources
- Line 173: a space is missing after Mthimkhulu & Aziakpono, 2015;
- Line 283: H3: Why the H3 hypothesis is not like H1 and H2 by focusing on women micro-entrepreneur businesses? It should be
o H3: There is a positive impact of Entrepreneurial Mentoring on the growth of women micro-entrepreneur businesses.
- Line 371: Provide more details regarding the 29 indicators utilized.
Author Response
- This research used a mixed-method with an explanatory design, this is a really good point. As mixed methods are not yet trendy, you may indicate a « mixed-method approach » somewhere in the title for a better paper discovery. I have improved the title: THE ROLE OF MICRO WAQF BANK IN WOMEN'S MICRO-BUSINESS EMPOWERMENT THROUGH ISLAMIC SOCIAL FINANCE: MIXED-METHOD EVIDENCE FROM MAWARIDUSSALAM INDONESIA.
- For un better understanding of the different amounts communicated, please provide the equivalence in USD. I have provided the equivalence in USD.
Specific remarks
- Lines 115-116: the same sentence appears twice: I have fixed in lIne 123-124
- Line 164: Human Resources replace by human resources: I have fixed in lIne 172
- Line 173: a space is missing after Mthimkhulu & Aziakpono, 2015; fixed at line 180
- Line 283: H3: Why the H3 hypothesis is not like H1 and H2 by focusing on women micro-entrepreneur businesses? It should be: H3: There is a positive impact of Entrepreneurial Mentoring on the growth of women micro-entrepreneur businesses. I have fixed at line 290.
- Line 371: Provide more details regarding the 29 indicators utilized: I have provided more detail at line 380-400
Reviewer 2 Report
The paper is well-written with an immaculate methodology. The results are presented well. Overall, good effort. There are following few comments for further clarity/improvement of the paper.
a. There is little clarity on what types of Shariah concepts & structures are adopted for offering the micro-financing products by MWB i.e. is it only loan (qard) based instruments or some other Shariah-compliant instruments that are used for the purpose?
b. Page 3 (line:115): Remove the repetition of the sentence "According to data, there are 62 MWB throughout Indonesia by 2022."
c. Page 12 (lines:476-78): Given the Shariah rule of complete ban on Riba (interest), further clarity is needed on how the margin of 3% is collected by MWB against the loan. Please justify
Author Response
a. I have clarified lines 581-583
b. I have removed the repetition lines 123-124
c. I have replaced capital loan with capital financing based on murabaha contract (485).
Reviewer 3 Report
There are some minor typos on lines 54, 111, 163, 165, 173, 181, 209. Please correct them.
Otherwise the paper is well written and research topic is relevant.
For theoretical discussion on Islamic Microlending (profit sharing) using joint responsibility system, cite Alistair Cameron, Mandar Oak, Yaping Shan, Peer monitoring and Islamic microfinance, Journal of Economic Behavior & Organization, Volume 184, 2021, Pages 337-358
Author Response
a. I have corrected typos on lines 54, 111, 163, 165, 173, 181, 209
b. For theoretical discussion on Islamic Microlending (profit sharing) using joint responsibility system, cite Alistair Cameron, Mandar Oak, Yaping Shan, Peer monitoring and Islamic microfinance, Journal of Economic Behavior & Organization, Volume 184, 2021, Pages 337-358. I have cited in reference Line: 232-233.
This manuscript is a resubmission of an earlier submission. The following is a list of the peer review reports and author responses from that submission.